# First-Principles Study on the Cu/Fe Interface Properties of Ternary Cu-Fe-X Alloys

**DOI:** 10.3390/ma13143112

**Published:** 2020-07-13

**Authors:** Yufei Wang, Min Li, Haiyan Gao, Jun Wang, Baode Sun

**Affiliations:** 1State Key Laboratory of Metal Matrix Composites, Shanghai Jiao Tong University, Shanghai 200240, China; yufeiwang@sjtu.edu.cn (Y.W.); lm2390393056@sjtu.edu.cn (M.L.); Junwang@sjtu.edu.cn (J.W.); bdsun@sjtu.edu.cn (B.S.); 2Shanghai Key Laboratory of Advanced High-Temperature Materials and Precision Forming, Shanghai Jiao Tong University, Shanghai 200240, China

**Keywords:** Cu/*γ*–Fe interface, interface properties, first-principles

## Abstract

The supersaturated Fe in Cu is known to reduce the electrical conductivity of Cu severely. However, the precipitation kinetics of Fe from Cu are sluggish. Alloying is one of the effective ways to accelerate the aging precipitation of Cu-Fe alloys. Nucleation plays an important role in the early stage of aging. The interface property of Cu/*γ*–Fe is a key parameter in understanding the nucleation mechanism of *γ*-Fe, which can be obviously affected with the addition of alloying elements. In this paper, first principles calculations were carried out to investigate the influence of alloying elements on the interface properties, including the geometric optimizations, interfacial energy, work of adhesion and electronic structure. Based on the previous research, 14 elements including B, Si, P, Al, Ge, S, Mg, Ag, Cd, Sn, In, Sb, Zr and Bi were selected for investigation. Results showed that all these alloying elements tend to concentrate in the Cu matrix with the specific substitution position of the atoms determined by the binding energy between Fe and alloy element (X). The bonding strength of the Cu/*γ*-Fe interface will decrease obviously after adding Ag, Mg and Cd, while a drop in interfacial energy of Cu/*γ*–Fe will happen when alloyed with Al, B, S, P, Si, Ge, Sn, Zr, Bi, Sb and In. Further study of the electronic structure found that Al and Zr were not effective alloying elements.

## 1. Introduction

Supersaturated Fe in a Cu matrix is known to reduce the electrical conductivity of Cu severely due to the slow precipitation kinetics of Fe [1,2,3]. Adding a third element allows a larger variety of possible kinetic paths during aging precipitation to improve the strength and conductivity of the binary Cu-Fe alloys [4,5]. The aging precipitation process of Cu-Fe alloy includes nucleation, growth and coarsening. However, few works focusing on the effect of the alloying elements on the nucleation of *γ*-Fe can be found. In the past, experimental investigations [6,7] of *γ*-Fe precipitates in Cu reveal that the antiferromagnetic behavior of *γ*-Fe precipitates at low temperatures has been well established. According to the order of magnetic moments of *γ*-Fe precipitates, the interface orientation relationship between *γ*-Fe precipitated and Cu matrix was (100)[010]Cu//(100)[010]*γ*–Fe, while confirming the absence of inter diffusion with surface alloy formation. The interface properties between the Cu matrix and *γ*-Fe, especially the atomic structure and interfacial energy, are crucial to make clear the nucleation mechanism of the *γ*-Fe in ternary Cu-Fe-X alloys. However, solid/solid interfacial adhesion and energy are difficult to be obtained by experiment. Recently, the first-principles calculation has been successfully used in the detailed study of metal/ceramic interface adhesion of Al-base [8], Mg [9] Fe [10] and Ni [11]. In the present work, the first principles calculation will be employed to study the Cu/*γ*-Fe interfacial properties to make clear the effect of alloying elements on the nucleation of *γ*-Fe in the early stage of aging of the Cu-Fe alloy. Based on the previous research [12,13], a large database of solute–vacancy binding energies in Cu was calculated, and then the solubility and diffusivities of iron in dilute Cu-Fe-X alloys were studied. These studies provide a basis for the selection of alloying elements; therefore, 14 elements including B, Si, P, Al, Ge, S, Mg, Ag, Cd, Sn, In, Sb, Zr and Bi will be selected for this investigation. Calculation on the substitutional energies, lattice misfits, work of adhesion, interfacial energy and electronic structure will be carried out with reasonably built Cu/*γ*-Fe interface configuration.

## 2. Models and Computation Method

Calculations were performed using the pseudopotential plane wave method within the density functional theory (*DFT*) *T* = 0 K as implemented in the *VASP* package [14,15,16,17]. The projector augmented wave (*PAW*) potentials were used with the generalized gradient approximation (*GGA*) [18,19,20,21] for exchange–correlation. The electron–ion interaction is described by the 3d, 4s states for Cu, the 3d, 4s states for Fe and the corresponding valence electron for the third elements. The integration over the Brillouin zone integrations was performed with a 6 × 6 × 1 Monkhorst–Pack *k*-point grid in case of interface and surface structures; a grid of 16 × 16 × 16 was used in case of bulk structures. An energy cutoff of 340 eV was used for the plane wave basis. The convergence of results with respect to energy cutoff and *k*-points was carefully considered. In order to obtain a precise crystal structure, atomic positions as well as unit cell volume and shape were fully relaxed. Ground-state geometries were determined by minimizing stresses and Hellmann–Feynman forces with the conjugate gradient algorithm, until the forces on all atomic sites were less than 10^−3^ eV/Å. Convergence of the electronic system was reached when the difference of the total energies between two iterations was below 10^−4^ eV. For magnetic 3d elements such as Fe, spin-polarized calculations were performed to analyze the influence of magnetic moments on the energy and electronic structure, *γ*–Fe was spin-polarized (antiferromagnetic ordering).

## 3. Results and Discussion

### 3.1. Bulk and Surface Calculation

The calculations on bulk properties of Cu and *γ*–Fe were carried out by *GGA*. The calculated lattice constant of Cu and *γ*–Fe were *a* = 3.634 Å and *a* = 3.447 Å, respectively, which were consistent with the experimental value and other calculation results. Table 1 shows the calculated lattice constants and cell volumes.

The free surfaces of the *γ*-Fe and Cu slabs with the in-plane periodicity were separated by a 15 Å vacuum to prohibit their interactions. Based on this model, the substitutional energies, lattice misfits, work of adhesion, interfacial energy and electronic structure were calculated, respectively. To make sure that both sides of the surface slabs of Cu (001) and *γ*–Fe (001) are thick enough to show the bulk-like interior, the convergence tests on Cu (001) and *γ*–Fe (001) surfaces with different thickness were carried out first. The calculated surface energies of Cu(100) and *γ*-Fe(100) slabs were obtained with the following equation [26,27]:(1)$$dE=TdS-PdV+\sigma dA+\sum_{i=1}^{C}\mu_{i}dn_{i}$$
where *E* is the total energy of the system, *T* is the temperature, *P* is the pressure, *V* is volume, σ is the surface energy, *A* is the surface area, *C* is the component fraction, $\mu_{i}$ is the chemical potential of the component and $n_{i}$ is the stoichiometry of the component. Because of the different substitutional sites of Cu/*γ*–Fe interface, either the X-occupied Fe site or the X-occupied Cu site, the chemical potentials were chosen carefully. Two possible reference states were considered: (1) a dilute solid solution Cu_n_X (*n* = 107); and (2) a dilute solid solution Fe_n_X (*n* = 107). For the X-occupied Cu site, the chemical potentials were calculated from the dilute solution Cu_107_X and pure Cu: $\mu_{X}=E_{{\mathrm{Cu}}_{107}X}-107E_{\mathrm{Cu}}$. For the X-occupied Fe site, the chemical potentials were calculated from the dilute solution Cu_107_X and pure Cu: $\mu_{X}=E_{{\mathrm{Fe}}_{107}X}-107E_{\mathrm{Fe}}$. Table 2 shows the calculated chemical potentials.

Scaled to 0 K, the σ was given by:(2)$$\sigma ={\left[\frac{\partial E}{\partial A}\right]}_{S,V,n_{i}}$$

For FCC Cu(100) and *γ*-Fe(100) slabs, the σ is defined as:(3)$$\sigma =\frac{1}{A_{surface}}\left(E_{slab}^{total}-n_{i}\mu_{i}\right)$$

$E_{slab}^{total}$ is the total energy of the surface model. In the calculated surface energies list in Table 3, it was found that the surface energies of the Cu(100) slabs and *γ*–Fe(100) slabs with more than six layers can converge to 1.48 J/m^2^ and 3.37 J/m^2^, respectively, which were in agreement with the experimental value and other calculated values.

### 3.2. Model Geometry of Interface

The interface properties are directly affected by the position of alloying elements. Therefore, the possible positions of alloy elements in Cu(100)/*γ*–Fe(100) interface were considered firstly in the calculations. The substitutional energy of alloy elements at different lattice positions in the interface model was calculated to determine the occupying tendency; the corresponding atomic position was shown in Figure 1.

The substitutional energy, $E_{\mathrm{Cu}\to X}^{fcc\;\mathrm{Cu}}$ in Cu and $E_{\mathrm{Fe}\to X}^{fcc\;\mathrm{Fe}}$ in Fe are defined as follows [6]:(4)$$E_{\mathrm{Cu}\to X}^{fcc\;\mathrm{Cu}}=\left(E_{{\mathrm{Cu}}_{n-1}{\mathrm{Fe}}_{n}X}^{total}+\mu_{\mathrm{Cu}}\right)-\left(E_{{\mathrm{Cu}}_{n}{\mathrm{Fe}}_{n}}^{total}+\mu_{X}\right)$$
(5)$$E_{\mathrm{Fe}\to X}^{fcc\;\mathrm{Fe}}=\left(E_{{\mathrm{Cu}}_{n}{\mathrm{Fe}}_{n-1}X}^{total}+\mu_{\mathrm{Fe}}\right)-\left(E_{{\mathrm{Cu}}_{n}{\mathrm{Fe}}_{n}}^{total}+\mu_{X}\right)$$
where $E_{}^{total}$ is the calculated total energy before and after substitution, *n* is the number of substitutional atoms and X is the alloy element. Table 4 shows the substitution energy of alloy elements at different lattice positions in the interface model.

The results showed that all of the alloying elements were concentrated in the Cu-based near the interface during the nucleation. The substitution positions of Ag and Mg atoms were concentrated in the 1 position and far away from the interface. The substitution position of the Cd atom was concentrated in the 3 position, which is the second nearest neighbor interface. The other atoms such as Al, B and Bi were concentrated in the 4 position, which is the nearest neighbor interface. The occupation law could be explained by the bonding strength between Fe and the alloy element (X). The calculation of the X–Fe binding energy studied in previous research [9] showed that the elements with strong X-Fe bonding were more likely to be concentrated near the interface. Therefore, ten atoms there (Al, B, Ge, In, P, S, Si, Zr, Bi and Sb) with a strong binding energy tended to be located in the nearest neighbor interface.

### 3.3. Work of Adhesion and Interfacial Energy

In the study of Finnis [29], the ideal work of adhesion *W_ad_* is defined as the bond energy needed to separate an interface into two free slabs; it is an important and convenient factor to predict the mechanical properties and the chemical bonding strength of an interface [30,31,32]. *W_ad_* was calculated by the following equation:(6)$$W_{ad}=\frac{1}{A_{interface}}\left(E_{\mathrm{Cu}}^{total}+E_{\mathrm{Fe}}^{total}-E_{\mathrm{Cu}/\gamma –\mathrm{Fe}}^{tatal}\right)$$
where $E_{\mathrm{Cu}/\gamma –\mathrm{Fe}}^{tatal}$ is the total energy of interface model, and $E_{\mathrm{Cu}}^{tatal}$ and $E_{\mathrm{Fe}}^{total}$ are the free surfaces energies of Cu and Fe, respectively.

Table 5 shows the effect of alloying elements on the work of adhesion *W_ad_*, the magnetic moment and the interfacial distance of fully relaxed interfaces.

The work of adhesion of binary Cu/*γ*-Fe is 3.822 J/m^2^, which is in agreement with the other calculated value, 3.89 J/m^2^ [20]. Furthermore, the work of adhesion of Cu/*γ*-Fe decreased by the addition of Ag, Bi, Cd, Ge, In, Mg, S, Sb and Sn, increased by the addition of Al, B, P, Si and Zr. Generally, lattice mismatch is an important reason for the change of work of adhesion, and the lattice mismatch at the interface of Cu/γ-Fe is determined by the volume of the alloy atoms. In order to get the influence of lattice misfits on work of adhesion, the volume of the solute atom $V_{Solute}^{X}$ was given by the volume difference induced by placing a single solute into pure Cu:(7)$$V_{Solute}^{X}=V\left({\mathrm{Cu}}_{n-1},X\right)-V\left({\mathrm{Cu}}_{n}\right)$$
where $V\left({\mathrm{Cu}}_{n-1},X\right)$ and $V\left({\mathrm{Cu}}_{n}\right)$ is the calculated volume before and after substitution. Then, the lattice misfit is given as:(8)$$\delta =\left(a_{\mathrm{Cu}}-a_{\gamma -\mathrm{Fe}}\right)/a_{\gamma -\mathrm{Fe}}$$
where *a* is the lattice constant.

The results list in Figure 2 shows that there was a positive correlation between the work of adhesion and the lattice misfits.

It means that the significant strain and lattice misfits can be reduced by adding relatively small atoms, thus, the work of adhesion of the interface becomes larger and the stable Cu/γ-Fe interface is formed. The lattice misfits at the interface of the Cu/γ-Fe effect by alloy elements vary from 5.3% to 6.1%, except for that of the Bi atom. However, the content of alloying elements is lower in reality, which indicates that the addition of alloying elements will not change the coherent relation of the Cu/γ-Fe interface. In addition, the magnetic moment and interfacial distance effect by alloying elements were given in Table 4. Under the influence of alloying elements, the magnetic moment of Fe varies from 2.367$\mu_{B}$ to 2.528$\mu_{B}$. The interfacial distance of binary Cu/*γ*-Fe is 1.83 Å, which is in agreement with another calculated value, 2.55 Å [20]. As the volume of alloying element increases, the interfacial distance increases, which should be due to the lattice distortion caused by the addition of a large atom. According to the thermodynamic theory of nucleation, interfacial energy as nucleation resistance is an important factor that hinders *γ*-Fe nucleation by influencing nucleation work and a critical nucleation radius. The effect of alloy elements on *γ*-Fe nucleation was determined by calculating the interfacial energy of Cu/*γ*–Fe. The interfacial energy can be given by:(9)$$\gamma =\frac{1}{A_{interface}}\left(E_{\mathrm{Cu}/\gamma –\mathrm{Fe}}^{tatal}-\overset{C}{\underset{i=1}{\sum }}\mu_{i}dn_{i}\right)-\sigma_{\mathrm{Cu}}-\sigma_{\mathrm{Fe}}$$
where *A* is the interface area, $E_{\mathrm{Cu}/\gamma –\mathrm{Fe}}^{tatal}$ is the total energy of a fully relaxed interface supercell, and *σ*_Cu_ and *σ*_Fe_ are the surface energy of the Cu(100) and *γ*-Fe(100), respectively. Table 6 shows the interfacial energy of Cu/*γ*-Fe effect by alloy elements.

The interfacial energy of Cu/*γ*-Fe in the binary Cu-Fe alloy is 0.819 J/m^2^. Furthermore, the interfacial energy of Cu/*γ*-Fe was enhanced with the addition of Ag, Mg and Cd. On the contrary, the interfacial energy decreased with the addition of the elements of Al, B, Bi, Ge, In, P, S, Sb, Si, Sn and Zr. It was known that the smaller the interfacial energy is, the more stable the interface structure is, the smaller the energy required to form the Cu/*γ*-Fe interface is, and then the more favorable to the nucleation of Fe it is.

### 3.4. Electronic Structure

The interface mechanical strength is closely related to the interfacial atomic bonding. In order to explore the interfacial bonding between Cu(100) and *γ*-Fe(100), the electron structure studies, including the difference charge density and the density of states, were calculated to obtain the bonding of atoms at the interface. The difference charge density Δρ was given by the following equation:(10)$$\Delta \rho =\rho_{\mathrm{Cu}/\gamma –\mathrm{Fe}}-\rho_{\mathrm{Cu}\left(100\right)}-\rho_{\gamma –\mathrm{Fe}\left(100\right)}$$
where $\rho_{\mathrm{Cu}/\gamma –\mathrm{Fe}}$ is the total charge density of Cu/*γ*–Fe interface, and $\rho_{\mathrm{Cu}\left(100\right)}$ and $\rho_{\gamma –\mathrm{Fe}\left(100\right)}$ are the charge densities of isolated Cu(100) and the Fe(100) surface, respectively.

Figure 3a displays the difference charge density for the Cu/*γ*–Fe interface; three typical elements with different relative positions were also given, as shown in Figure 3 The figures show the charge density difference between the Cu/*γ*–Fe system and that of the $\rho_{\gamma –\mathrm{Fe}\left(100\right)}$ system and the $\rho_{\gamma –\mathrm{Fe}\left(100\right)}$ system. It is obvious that the perturbation created by the Fe-Cu binding was mostly localized between Cu and *γ*–Fe interface, which indicates that the bonding is characterized by covalent and will not be changed by the addition of the third elements. However, the binding strength of Cu/*γ*–Fe interface was significantly changed by the addition of the third elements. These variations of the binding strength were mainly caused by charge transfer. It is found that the charge density at the interface decreases obviously and the bonding between atoms decreases with the addition of alloy elements Ag, Cd and Mg, as illustrated in Figure 3b. The charge density at the interface increases slightly and the bonding between atoms increases with the addition of alloy elements B, S, Si, Zr, Al, Bi, Ge, In, P, Sb and Sn, as illustrated in Figure 3c. 

To investigate the electron population and bonding strength, local density of states (LDOS) only containing the atoms near the interface was further calculated. Figure 4 shows the LDOS of Cu/*γ*–Fe interface, the difference Δ*N* between the LDOS of Cu/*γ*–Fe system and that of Cu(100) system and Fe(100) system revealed clearly the Fe-Cu bonding and antibonding states, which were just below and above the Fermi level: bonding states were around −5.43~−3.54 eV below and antibonding states were around 1.64~3.25 eV above. For LDOS, the energy range from −3.54 to 1.64 eV and the peaks of the Cu/*γ*–Fe system coincide with the peaks of LDOS in the inner shell of Cu and *γ*–Fe, which means that they are atomic ground state electrons. The positive and negative fluctuations in the Δ*N* indicate electron migration. For the bonding states, which were mainly composed of 3d states of Cu and 3d states of Fe, one peak appeared at −4.21 eV. Obviously, a covalent band formed at this coverage and the calculated charge number of the bonding states is 10.79 (charge numbers are the result of integration of the curves).

For ternary alloy, Cu-Fe-Ag(Si) were taken as an example; the Fe-Cu bonding and antibonding states were determined by the difference Δ*N* between the LDOS of the Cu-Ag(Si)/*γ*–Fe system and that of the Cu-Ag(Si) (100) system and Fe(100) system. The Cu-Ag/*γ*–Fe interface list in Figure 5 shows that the bonding states were around −5.45~−3.51 eV above the Fermi level and the antibonding states were around 1.69~3.29 eV.

Compared with the LDOS analysis of the binary Cu/*γ*–Fe interface, in the ternary Cu-Ag/*γ*–Fe interface, the calculated charge number of the bonding states decreased from 10.79 to 10.00, indicating that the less charge transferred from Fe (Cu) to bonding states and the bonding strength decreases.

For the Cu-Si/*γ*–Fe interface list in Figure 6, the bonding states around −7.67~−3.61 eV was above the Fermi level and antibonding states were around 1.62~3.20 eV.

Compared with the LDOS analysis of the binary Cu/*γ*–Fe interface, in the ternary Cu-Si/*γ*–Fe interface, the calculated charge number of the bonding states increased from 10.79 to 12.10, indicating that more charge was transferred from Fe (Cu) to bonding states and that the bonding strength increased. More computational results about the charge number of the bonding states affected by alloying elements were listed in Table 7. The calculation revealed that 5 elements Al, Ag, Mg, Cd and Zr can reduce the charge number of the bonding states, which suggests that the bonding strength will decrease by the addition of these elements. The weakening of interatomic bond at the interface indicates that the higher the energy required to form the interface, the higher the interface energy is.

In summary, according to the analysis of interfacial energy, eleven atoms Al, B, Bi, Ge, In, P, S, Sb, Si, Sn and Zr decrease the interfacial energy, and the addition of large solute atoms increases the interfacial distance. However, by studying the electronic structure, it was found that the interfacial distance should be due to the lattice distortion caused by the addition of a large atom. Although the interfacial distance increases, because the solute atoms participated in the bonding at the interface, the charge numbers of the bonding states were enhanced and the interface energies were reduced. Therefore, it was found that nine alloy elements will promote the precipitation of *γ*-Fe and form stable *γ*-Fe phases during nucleation.

## 4. Conclusions

The influence of the nucleation of Fe was investigated by studying the Cu/*γ*–Fe interfacial properties. The substitutional energy of alloy elements at different lattice positions of the interface were calculated to determine the occupying tendency. The results showed that the all of the alloying elements were concentrated in the Cu-based, the substitution position of Ag and Mg atoms are concentrated in the most far away from the interface, Cd atom is concentrated in the secondary adjacent interface position, and the rest of the atoms were concentrated in the nearest neighbor interface. Under the condition of coherent interface, the interfacial energy was dominated by the work of adhesion, so the distribution of interfacial energy was proportional to the size of atoms. However, under the influence of the bonding strength at the interface, 3 alloy elements—Ag, Mg and Cd—obviously increase the interface energy. The addition of 11 elements—Al, B, S, P, Si, Ge, Sn, Zr, Bi, Sb and In—can reduce the interfacial energy of Cu/*γ* –Fe, while among those, Al and Zr can reduce the charge number of the bonding states. Therefore, these nine alloy elements will promote the precipitation of *γ*-Fe and form stable *γ*-Fe phases during nucleation.

## Figures and Tables

**Figure 1 materials-13-03112-f001:**
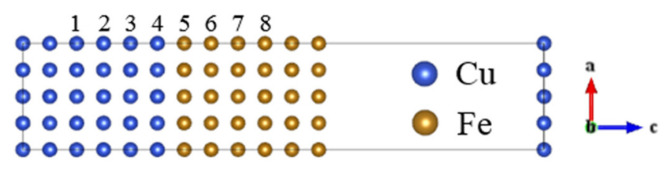
Structure of Cu(100)/*γ*–Fe(100) interface. The different layers of the interface area are marked with 1–8.

**Figure 2 materials-13-03112-f002:**
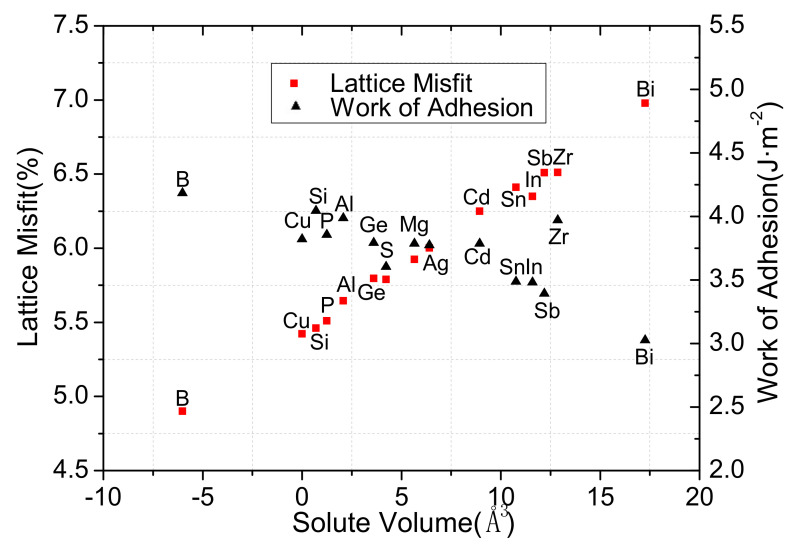
Work of adhesion and lattice misfit of Cu/*γ*–Fe interface as a function of solute volume.

**Figure 3 materials-13-03112-f003:**
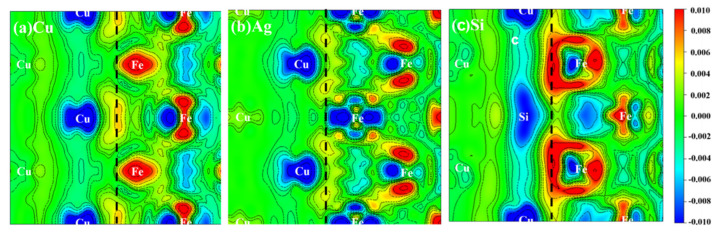
Difference charge density (e/Å^3^) of Cu/*γ*–Fe interface. (**a**–**c**) represent no solute atom, a solute atom Ag and Si, respectively. The dashed lines indicate the location of the interfaces. Areas of electron accumulation (red) and depletion (blue) have positive and negative signs, respectively; contour lines are drawn at 0.002 e/Å^3^ intervals.

**Figure 4 materials-13-03112-f004:**
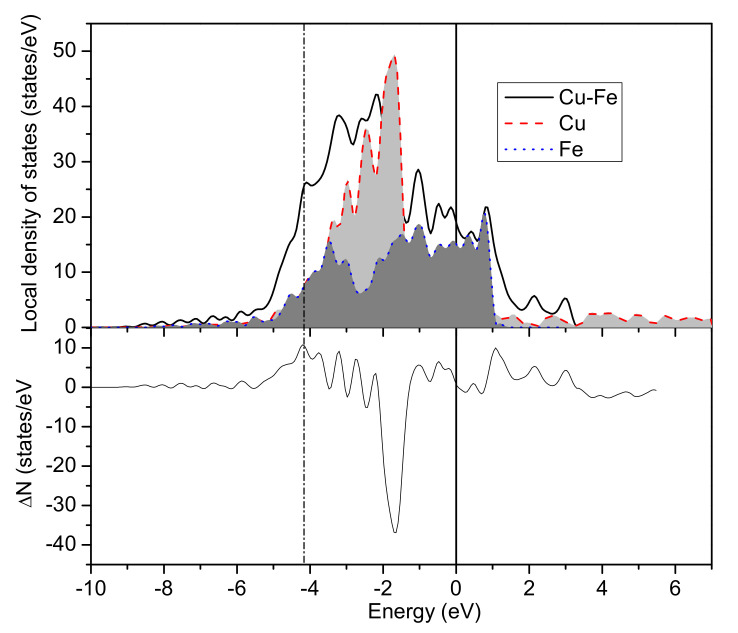
The LDOS of the binary Cu-Fe system, the Fermi energy is set to 0 eV. Δ*N* is the difference between the LDOS of Cu/*γ*–Fe system and that of Cu(100) system and Fe(100) system.

**Figure 5 materials-13-03112-f005:**
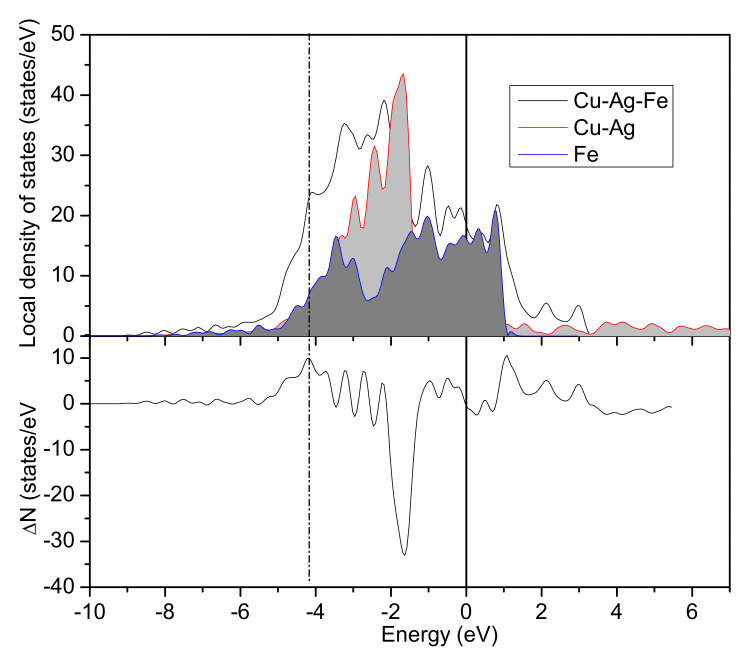
The LDOS of the binary Cu-Fe-Ag system; the Fermi energy is set to 0 eV. Δ*N* is the difference between the LDOS of the Cu-Ag/*γ*–Fe system and that of the Cu-Ag(100) system and the Fe(100) system.

**Figure 6 materials-13-03112-f006:**
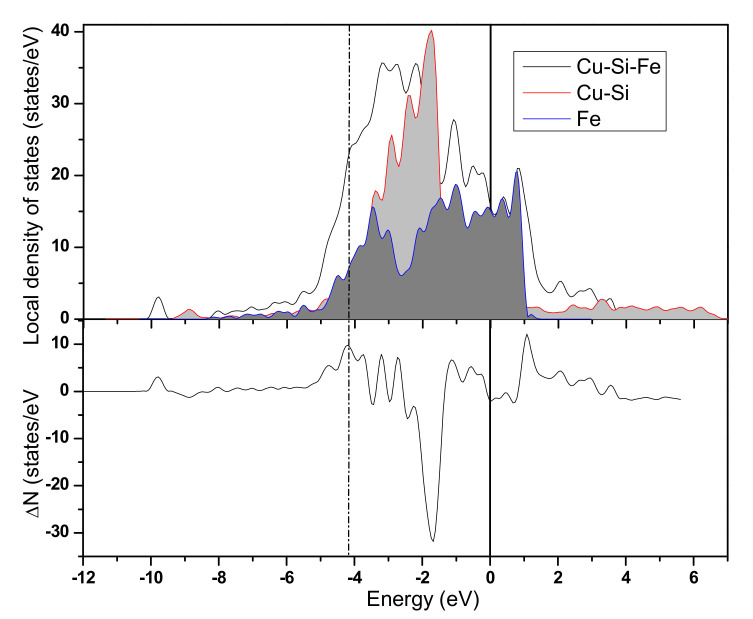
The LDOS of the binary Cu-Fe-Si system; the Fermi energy is set to 0 eV. Δ*N* is the difference between the LDOS of Cu-Si/*γ*–Fe system and that of Cu-Si(100) system and Fe(100) system.

**Table 1 materials-13-03112-t001:** The lattice constants and cell volumes.

Parameter	Cu				*γ*–Fe			
	*GGA* (this work)	*GGA* [18]	*GGA* [22]	Exp. [23]	*GGA* (this work)	*GGA-PBE* [24]	*GGA-PW91* [24]	Exp. [25]
a/Å	3.63	3.63	3.64	3.62	3.45	3.47	3.47	3.65
*V*/Å^3^·cell^−1^	48.00	47.99	48.23	47.24	40.95	41.93	41.85	48.79

**Table 2 materials-13-03112-t002:** Chemical potentials of elements (X), a dilute solid solution Cu_107_X and a dilute solid solution Fe_107_X.

Elements (X)	$\mu_{X}(\mathrm{eV})$	
	Cu_107_X	Fe_107_X
B	4.802	7.103
Si	5.659	7.235
P	5.258	6.595
Al	4.447	4.691
Ge	4.514	5.605
S	3.529	4.261
Mg	1.73	0.911
Ag	2.294	1.296
Cd	0.32	0.44
Sn	3.56	3.416
In	2.286	1.637
Sb	3.618	3.742
Zr	8.143	8.071
Bi	2.459	1.877

**Table 3 materials-13-03112-t003:** Surface energies of Cu(100) and *γ*–Fe(100).

Layer Number/*n*	*σ*/J·m^−2^	
	Cu(100)	*γ*–Fe(100)
4	1.45	3.30
6	1.48	3.36
8	1.48	3.37
10	1.47	3.36
12	1.46	3.38
Ref	1.485, 1.532 [22] 1.79 J/m^2^ [28]	2.956, 2.973 [14]

**Table 4 materials-13-03112-t004:** Substitutional energies (eV) of alloy atoms at different positions. Figure 1 lists the 1–8 representing the different substitutional sites. Underline indicates the smallest substitutional energy.

Elements	1	2	3	4	5	6	7	8
B	1.431	1.439	1.751	0.188	0.276	0.663	0.616	0.666
Si	−0.240	−0.284	−0.312	−1.165	−1.076	−0.928	−1.099	−1.052
P	0.107	0.033	0.010	−1.065	−0.954	−0.784	−0.953	−0.915
Al	−0.763	−0.777	−0.792	−1.235	−0.817	−0.414	−0.621	−0.585
Ge	0.084	0.039	0.012	−0.673	−0.384	−0.038	−0.251	−0.235
S	0.635	0.535	0.769	−0.544	0.048	0.696	0.497	0.516
Mg	−0.121	−0.119	−0.111	0.000	0.685	1.438	1.171	1.177
Ag	0.595	0.602	0.604	0.744	1.564	2.450	2.293	2.258
Cd	0.649	0.648	0.642	0.643	1.536	2.503	2.212	2.176
Sn	0.518	0.476	0.459	−0.050	0.573	1.225	0.854	0.862
In	0.503	0.484	0.484	0.255	1.042	1.877	1.538	1.514
Sb	0.878	0.800	0.766	0.063	0.564	1.087	0.740	0.727
Zr	0.536	0.453	0.495	−0.336	0.328	0.966	0.598	0.620
Bi	1.916	1.844	1.818	1.349	2.055	2.863	2.409	2.395

**Table 5 materials-13-03112-t005:** Effects of alloying elements on the work of adhesion, the magnetic moment and the interfacial distance of fully relaxed interfaces.

Elements	$W_{ad}(J\cdot m^{-2})$	$\mathrm{Magnetic}\;\mathrm{Moment}\;(\mu_{B})$	Interfacial Distance (Å)
Cu	3.822	2.441	1.827
B	4.185	2.409	1.827
Si	4.045	2.418	1.825
P	3.856	2.477	1.827
Al	3.988	2.452	1.831
Ge	3.793	2.367	1.832
S	3.604	2.406	1.838
Mg	3.786	2.479	1.827
Ag	3.775	2.502	1.827
Cd	3.786	2.451	1.837
Sn	3.488	2.507	1.847
In	3.481	2.510	1.849
Sb	3.394	2.494	1.847
Zr	3.972	2.416	1.844
Bi	3.027	2.528	1.855

**Table 6 materials-13-03112-t006:** Effects of alloying elements on interfacial energies of Cu/*γ*–Fe interface.

Element	Cu	Ag	Al	B	Bi	Cd	Ge	In
γ/J·m^−2^	0.819	0.852	0.659	0.288	0.750	0.869	0.582	0.641
**Element**	**Mg**	**P**	**S**	**Sb**	**Si**	**Sn**	**Zr**	
γ/J·m^−2^	0.845	0.443	0.474	0.584	0.503	0.684	0.579	

**Table 7 materials-13-03112-t007:** The charge number of the bonding states affected by alloying elements.

Element	Cu	Ag	Al	B	Bi	Cd	Ge	In
	10.79	10.00	9.71	10.92	12.84	10.69	13.27	11.94
**Element**	**Mg**	**P**	**S**	**Sb**	**Si**	**Sn**	**Zr**	
	9.74	12.40	12.17	13.12	12.10	12.62	8.18

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
