# Peer review of "First-Principles Study on the Cu/Fe Interface Properties of Ternary Cu-Fe-X Alloys"

_materials, 2020, doi:10.3390/ma13143112_

Round 1
Reviewer 1 Report
This paper presents first principles calculations carried out to investigate the influence of alloying elements on the Cu/gamma-Fe interface properties.
The method shold be described more precisely:
(i) the choice of the computation cell is not explained. The authors have chosen to built a supercell with 3 interfaces (Cu/void, Fe/void, Cu/Fe). They should justify that there is no interference between the three interfaces, which is not clear for me. The authors give a reference to explain why they chose the (100) interfacial plane. However, this paper does not mention explicitely the (100) plane. The choice of the (111) interfacial plane would have been more straighforward, because it corresponds to the densest plane in the respective bulk metals. A explanation is required here.
(ii) the computational details should be given (k-point grid, spin polarization, cell shape and volume fixed or not, convergence criteria for the electronic structure calculations, etc).
(iii) The method to calculate the binding energy is not detailed
(iv) The method to calculate V_X^solute is not clear. Have the authors performed bulk calculations?
(v) page 2, the authors assumed that their value for the surface energy (1.48 J/m2) is in agreement with the experimental value (0.82 J/m2). The agreement is in fact really poor.
The study uses the chemical potentials of the considered metals. Several details are missing:
(i) The values for mu_i in bulk i are not given. They should be compared with the ones from experiment (cohesive energies).
(ii) The values for mu_i in another metal j may differ from the previous ones. How has it been taken into account in the calculations?
Electronic structure calculations are questionnable:
(i) the authors did DOS calculations. However, they talk about bonding and antibonding states, which requires the calculation of COHPs.
(ii) Numbers given on page 7 to describe fig. 4 seems not to be in agreement with the numbers of Fig. 4
The conclusion of the authors given in Fig. 8 is questionnable: the linear fit is very poor and does not show any clear correlation between the interfacial energy and the solute volume.
Additional comments
(i) Ref 22 is not written in English
(ii) the sentences should be more rigorous. For exemple, "most advantageous" is not clear. The authors should use the appropriate scientific words
(iii) Fig1 (a) and (b) are the same.
(iv) Fig. 3 : units are not given
(v) Fig. 4, Fig. 5: the LDOS for Cu is hidden in the region [-2 eV; 0 eV]. The vertical lines are not explained. The plot in the lower pannel is not explained.
(vi) Tab 4 : NN is not defined
Author Response
(1)According to the order of magnetic moments of γ-Fe precipitates, experimental investigations reveal that the interface orientation relationship between γ-Fe precipitated and Cu matrix was (100)[010]Cu//(100)[010]γ–Fe. Therefore, the same interface orientation relationship was used in this paper. The corresponding explanation was added to the introduction.
(2) As the referee’s suggestions, the more computational details were given.
(3) Due to copyright issues in Table 4, Table 4 has been deleted and the corresponding number has been revised.
(4) An error experimental value (0.82 J/m2) has been fixed.
(5) As the referee’s suggestions, a detailed explanation of the chemical formula was added to the manuscript.
(6) As the referee pointed out, COHP were usually used to study bonding and antibonding states. But the difference between the LDOS of Cu/γ–Fe system and that of Cu(100) system and Fe(100) system can also revealed clearly the Fe-Cu bonding and anti-bonding states. This method can be found in (J. Phys. Chem. B 108 (2004) 14477. in page 5,). More detailed explanation of charge number was added.
(7) As the referee’s suggestions, Fig.1 and Fig.3 have been revised. Fig.7 and Fig.8 have been deleted, and the corresponding number have been revised. The content in Figure 7 was rephrased to Table 6.
(8) Some typos errors have been corrected and some poorly organized sentences have been reorganized.
Reviewer 2 Report
Wang and coworkers have explored FCC Cu/Fe interfaces with additions of 14 alloying elements using density functional theory. The interfacial properties are of relevance for nucleation of FCC Fe precipitants in FCC Cu, altering the electrical conductivity. The authors have proposed the electronic structure origin of the interface energetics. This work is interesting, but there are many issues to be addressed. These are listed below.
1) FCC Fe exhibits AFM ordering (collinear and non-collinear configurations are possible). The authors have either ignored magnetism or didn’t discuss it. If the magnetic effects were ignored, the manuscript should be rejected (some indication for this case is that lattice parameter and surface energy of Fe are considerably worse reproduced that those of Cu). If the authors simply didn’t discuss it, they should do it in the revision. How does magnetic moment change as a function of distance to the interface? How is the electronic structure affected?
2) The authors should motive as to why (001) planes were considered and not (111).
3) More details are required for the alloying element selection. Are the corresponding interfaces coherent? How do the current findings correspond to the previously published data? More discussions are required.
4) The authors should review the literature more carefully and compare their data with the available reports (e.g. J. Magn. Magn. Mater. 321, 2260 (2020); Metals 8, 384 (2018); etc.).
5) More details are required in the methodological section (k-points, energy convergence, spin polarization, etc.).
6) The use of italics in not consistent (variables are in italics in the equations, but nearly never in the text).
7) Based on Fig. 2 (lattice mismatch), the explored Cu/Fe interfaces may be semi-coherent or even incoherent. At least, misfit dislocations should form. The authors should discuss the Cu/Fe more realistically even though their DFT data are highly idealized. A good discussion is always helpful.
8) The Cu/Fe system should also exhibit metallic bonding, which were completely ignored by the authors. This is not acceptable.
9) The electronic structure is analyzed for the pristine Cu/Fe interface and with Ag, Al, and Si additions in Fig. 3, but in Fig. 6 the B addition (instead of Si) is further explored. This gives rise to confusion and is hence not acceptable. Please always use the same examples.
10) There is no evidence that anti-bonding occurs at the proposed energy range in Fig. 4, Fig. 5, and Fig. 6. Empty states are not necessary anti-bonding. To prove this, more analysis is required (e.g. COOP, COHP, etc.).
11) It is not clear what “N (states/eV” stands for in Fig. 4, Fig. 5, and Fig. 6. Please elaborate.
12) In Fig. 6, the states at -8.88 eV are core states and have nothing to do with the overall bonding (valence states).
13) The alloying elements should be logically ordered (e.g. increasing atomic number). The current sequence is chaotic and especially confusing in Fig. 7. The author should also define the “charge number”. For instance, B has 3 valence electrons. How is this “charge number” 10.92 for the B containing system?
14) Fig. 8 is used to show that there is a correlation between the interfacial energy and volume. This is incorrect. There’s actually no correlation (e.g. Ge and Zr give rise to a drastically different volume, but the interfacial energy is nearly identical). Please remove this figure or rephrase the text.
15) There’s a lot of confusion regarding different treatment of energetics (interfacial energy, work of adhesion, and substitutional energy). The authors need to decide what description is the most appropriate and either use it to explain the reported experimental findings or use it to predict the new ones. How are these results relevant? What should experimentalists learn from this work?
16) English should be improved. There are grammatical mistakes (e.g. “Table 3 shown the substitutional energy”), many odd terms are used (e.g. “public literatures”), and there are contradictions (e.g. Al increases and decreases the bonding strength – two sentences in the abstract and similar contradictions in the conclusions). Furthermore, Ref. 22 is not even in English.
Author Response
(1) For magnetic 3d elements Fe, spin-polarized calculations were performed to analyze the influence of magnetic moments on the energy and electronic structure. The main purpose of this paper is to study the effect of alloying elements on interfacial energy and interfacial bonding strength. More investigation on how does magnetic moment change as a function of distance to the interface and electronic structure, we would like to discuss it in further studies.
(2)According to the order of magnetic moments of γ-Fe precipitates, experimental investigations reveal that the interface orientation relationship between γ-Fe precipitated and Cu matrix was (100)[010]Cu//(100)[010]γ–Fe. Therefore, the same interface orientation relationship was used in this paper. The corresponding explanation was added to the introduction.
(3) As the referee’s suggestions, more details of alloying element selection were added.
(4) As the referee’s suggestions, the more computational details were given.
(5) The equations have been revised in the text.
(6) In order to obtain a precise interface structure, atomic positions as well as unit-cell volume and shape were fully relaxed. The effect of low alloy addition on lattice mismatch is very limited.
(7) As the referee’s suggestions, Fig.3 and Fig.6 has been revised. Fig.6 discussed the Si addition instead of B. The explanation of â–³N was defined in the text.
(8) As the referee pointed out, COHP were usually used to study bonding and antibonding states. But the difference between the LDOS of Cu/γ–Fe system and that of Cu(100) system and Fe(100) system can also revealed clearly the Fe-Cu bonding and anti-bonding states. This method can be found in (J. Phys. Chem. B 108 (2004) 14477. in page 5,). More detailed explanation of charge number was added.
(9) As the referee’s suggestions, Fig.7 and Fig.8 has been deleted, and the corresponding number has been revised. The content in Figure 7 was rephrased to Table 6.
(10) Some typos errors have been corrected and some poorly organized sentences have been reorganized.
Round 2
Reviewer 1 Report
The reply from the authors does not take into account all comments.
1- page 3 "Because of the different substitutional sites of Cu/γ–Fe interface, either X occupy Fe site or X occupy Cu site, the chemical potentials were chosen carefully."
X is not defined at this stage.
2- The equation ?? = ???107? − 107?? is not understandable. You have to label it. You should mean
?? = ???107? − 107E??.
3- The method to calculate the binding energy is not detailed
It should be detailed.
5- The autors gave a value for the surface energy of Cu(100) which is not the experimental value, while they assume that it is an experimental valu in the text. For experimental values, see the paper by Vitos.
The authors should also give the surface energy of gamma-Fe and the corresponding reference
6- The values for mu_i in bulk i are not given. They should be compared with the ones from experiment (cohesive energies).
7- The values for mu_i in another metal j may differ from the previous ones. They should appear in the text.
Author Response
1) The formula of has been revised. The definition of X was added.
(2) Since the relevant data about the calculation of binding energy were deleted in this manuscript, the calculation method of binding energy was not discussed.
(3) As the referee’s suggestions, experimental values from Vitos were referenced in this manuscript.
(4) The value of cohesive energies in the two phases is not the focus of this paper. The authors hope to show more valuable information in the manuscript. Therefore, we hope to present only the calculation method in this article.
(5) Some typos errors have been corrected and some poorly organized sentences have been reorganized.
Reviewer 2 Report
The authors have improved the manuscript to some extent, but there are still open issues.
1) Claiming spin polarization without any specification (is it FM, AFM-Q1, AFM-Q2, AFM-Q3, …?) and discussion (magnetic moments) as well as ignoring it in the electronic structure analysis (Fig. 4, Fig. 5, Fig. 6) is not acceptable.
2) The authors have refused to compare their data with the available literature (e.g. J. Magn. Magn. Mater. 321, 2260 (2020); Metals 8, 384 (2018); etc.). This is not acceptable.
3) The use of italics is still inconsistent (equations vs. text, see e.g. page 2).
4) Lattice mismatch up to 7% may easily lead to incoherency, but the authors have refused to address/discuss this issue. This is not acceptable.
5) Delta N is discussed in the text, but N is used in Fig. 4, Fig. 5, and Fig. 6. Furthermore, delta N should be better described (the meaning of positive and negative values).
6) More discussions are required. How are these results relevant? What should experimentalists learn from this work?
7) English should be improved. There are many errors (e.g. in the abstract “Interface properties … is a key parameter…”, “Results shown that…”, etc.). Furthermore, there are logical mistakes (e.g. in the abstract: “Bonding strength of … will decrease obviously after adding Al…” so Al decreases the bonding strength vs. “However, only five alloy elements Al… are effective alloying elements due to the strongly bonded … interfaces.” so Al increases the bonding strength).
Author Response
(1)More investigation of magnetic moment and interfacial distance were added to Table 4, and the corresponding discussion has also been added in the text. The influence of electron spin was fully considered in the study of electronic structure. The LDOS given in figure 4-6 were the total density of states containing both of up spin and down spin.
(2) As the referee’s suggestions, the values from Metals, 8 (2018) 384 were referenced in this manuscript.
(3) The use of italics were further corrected.
(4) As the referee referred to, high lattice mismatches tend to lead to inconsistencies, which is a very good suggestion. But the lattice misfits at the interface of Cu/ γ-Fe effect by alloy elements varies from 5.3% to 6.1%, except for Bi atom. However, the content of alloying elements is more lower in reality, which indicates that the addition of alloying elements will not change the coherent relation of the Cu/ γ -Fe interface. The corresponding discussion was added to the text.
(5) As the referee’s suggestions, more details of â–³N were given.
(6) More discussion about the content, details, and significance of this article has been added.
(7) Some typos errors have been corrected and some poorly organized sentences have been reorganized.
Round 3
Reviewer 1 Report
(4) The value of cohesive energies in the two phases is not the focus of this paper. The authors hope to show more valuable information in the manuscript. Therefore, we hope to present only the calculation method in this article.
Surface energies, interfacial energies, substitutional energies, depend on the chemical potential. These energies are valuable results detailed by the authors. It seems to me that the values for the chemical potentials, used by the authors to produce their valuable results, should be mentioned in the paper. It will ensure the reproductibility of the results by other groups.
Author Response
(1) As the referee’s suggestions, the calculated chemical potentials were listed in table 2. New table 2 was added and the corresponding number has been revised.
(2) Some typos errors have been corrected and some poorly organized sentences have been reorganized.
Reviewer 2 Report
The authors have again improved the manuscript to some extent, but there are still open issues.
1) Please STATE what kind of spin polarization you used. The manuscript cannot be accepted without this information.
2) The use of italics is inconsistent (equations vs. text, see e.g. lattice parameter “a” on page 2).
3) Delta N is discussed in the text, but N is used in Fig. 4, Fig. 5, and Fig. 6. Please fix these figures!!!
4) English should be improved. There are still many errors in the manuscript (e.g. in the abstract “Interface properties … is …”
Author Response
(1) As the referee’s suggestions, the kind of spin polarization was added in the computation method.
(2) The use of italics were further corrected.
(3) As the referee’s suggestions, more details of â–³N were added to fig 4, fig 5 and fig 6.
(4) Some typos errors have been corrected and some poorly organized sentences have been reorganized.
Round 4
Reviewer 2 Report
The manuscript is acceptable in the current form. However, there’s still a technical issue with Fig. 4, Fig. 5, and Fig. 6. Delta N (symbol delta) is not visible in these figures (at least not in the available pdf file). This should be fixed before publication.